# Electropolishing Stainless Steel Optimization Using Surface Quality, Dimensional Accuracy, and Electrical Consumption Criteria

**DOI:** 10.3390/ma16051770

**Published:** 2023-02-21

**Authors:** Elena María Beamud-González, Pedro José Núñez-López, Eustaquio García-Plaza

**Affiliations:** 1Mining and Industrial Engineering School, Department Applied Mechanics & Project Engineering, University of Castilla-La Mancha, Plaza Manuel Meca 1, 13400 Almadén, Spain; 2Higher Technical School of Industrial Engineering, Energy Research and Industrial Applications Institute (INEI), Department Applied Mechanics & Project Engineering, University of Castilla-La Mancha (UCLM), Avda. Camilo José Cela, s/n, 13071 Ciudad Real, Spain

**Keywords:** individual and multi-objective optimization, electropolishing, polishing rate, surface roughness, dimensional accuracy, electrical consumption cost

## Abstract

Electropolishing (EP) processes require high electrical consumption that must be optimized to minimize production costs without sacrificing the objectives of surface quality and dimensional accuracy. The aim of the present paper was to analyze the effects of the interelectrode gap, initial surface roughness, electrolyte temperature, current density, and EP time on aspects of the EP process applied to AISI 316L stainless steel, which have not been examined in the literature, such as polishing rate, final surface roughness, dimensional accuracy, and electrical consumption cost. In addition, the paper aimed to obtain optimum individual and multi-objective considering criteria of surface quality, dimensional accuracy, and electrical consumption cost. The results showed that the electrode gap was not significant on the surface finish or current density, and the EP time was the parameter having the greatest effect on all criteria analyzed, with a temperature of 35 °C showing the best electrolyte performance. The initial surface texture with the lowest roughness ${Ra}_{1}^{0}$ (0.5 ≤ *Ra* ≤ 0.8 μm) obtained the best results with a maximum polishing rate of ~90% and minimum final roughness (*Ra*) of ~0.035 μm. The response surface methodology showed the EP parameter effects and the optimum individual objective. The desirability function obtained the best global multi-objective optimum, while the overlapping contour plot provided optimum individual and simultaneous per polishing range.

## 1. Introduction

The performance of the electropolishing (EP) process is conditioned by several factors such as current density, electropolishing time, electrolyte temperature and concentration, electrode gap, the distance between electrodes, initial surface finish, workpiece and electrode material, among others. The electrolyte affects electrical conductivity and chemical reactivity, which in turn determines the performance of the process in terms of surface finish [1], smoothness [2], and gloss [3]. The most extensively used electrolytes employed with stainless steel contain sulphuric acid, phosphoric acid, a mixture of both, or occasionally mixed with complementary additives such as glycerol or chromic acid [4]. Though additives such as glycerol have been used to improve specific process properties [5,6], both of these studies concluded that improved surface finish was achieved through a combination of a set of temperature ranges and current densities. Moreover, the addition of chromic acid to the electrolyte improved surface gloss [3], but no significant effects on surface smoothness [2] or roughness [1] were found, with the added drawback of high toxicity. Other studies analyzing the effects of organic additives on the performance of the EP process, where amines improved process performance as compared to glycerol by reducing application times [7], found that the anodic corrosion rate increased by raising the temperature for the different amine concentrations [8].

The optimization of EP conditions has been extensively examined in the literature, which has tended to assess the performance of the EP process, and a wide array of factors, with most studies focusing on electrolyte temperature, EP time, and current density. The temperatures that have been analyzed have ranged from −70 °C to 75 °C, with most studies falling within the 35 °C to 75 °C temperature range. Optimum electrolyte temperatures have been obtained for titanium and niobium at −70 °C [9], aluminum at 75 °C [10], copper at 65 °C [11], and stainless steel at 35 °C [1]. EP time is another crucial parameter, with optimum surface gloss values for stainless steel at 10 min [12] and 25 min for aluminum [13], but excessive EP time worsens surface roughness [14], leading to the reappearance of irregularities that had disappeared at a short interval (four min), with a longer EP time of eight min due to the formation of chromic carbon [15]. Finally, the current density is the most widely evaluated parameter in the literature owing to its impact on gloss [3], smoothness [2], surface roughness [16], and other material properties, i.e., thickness, elastic modulus, and hardness [17]. Alternatively, several authors have examined other process variables such as the distance between electrodes [18], surface gloss [3], electrolyte concentration [11], viscosity [19], agitation [20], magnetic field [21], salinity [22], and rotating disk electrodes [23], among others.

Surface quality in terms of roughness, gloss, and smoothness is one of the key EP processes proprieties that has been extensively assessed. Several studies analyzing surface quality by combining shot peening with electropolishing have found it considerably improved the surface finish [24], whilst other studies found surface roughness and gloss improved by combining a magnetic field with the EP process in 316L stainless steel [25]. Other authors have assessed hardness, corrosion resistance, or surface hydrophobicity, where an increase in electrical current combined with a eutectic mixture of choline chloride and ethylene glycol improved these variables in materials such as aluminum [26]. Moreover, corrosion has been evaluated in a broad spectrum of materials such as copper, niobium, nickel and titanium alloys, stainless steel [27], stainless steel protection against bacterial colonization [22], surface passivation at high current densities [28], surface optimization for the adhesion of plasma-deposited film [29], and in medical implants and devices [21]. Furthermore, the effects of electropolishing on thermal coated materials in either spray or overlapping layers obtained with a rotating electrode have been compared [30], as well as the effect on nickel-titanium memory actuator sheets, where a reduction in internal roughness lowered energy requirements and increased the heat transference ratio [31]. Finally, cleaning and disinfecting processes of stainless steel surfaces combining electropolishing with alkaline water were found to eliminate more microorganisms at shorter exposure times [32].

The main drawback of EP processes is non-homogeneous material elimination. The entire surface of the product in contact with the electrolyte is affected, and the elimination of material is conditioned by the distance between the electrodes and the treated surface texture. This may lead to dimensional and geometric product modifications that have a substantial impact on the design tolerances, assembly, performance, and service life. Moreover, electrical energy consumption is a critical factor to be borne in mind in the selection of EP conditions, as it significantly determines the cost of the final product. Therefore, the quality of electropolished products depends on multiple criteria determined by the requirements of manufacturers and consumers. In industrial production, the simultaneous optimization [33] of process factors is standard practice, and product quality and cost are the factors most extensively analyzed.

The characterization and optimization of the EP process applied to improve the surface quality of components manufactured with AISI 316L stainless steel is a crucial aspect in the industrial sector. This stainless steel is widely used in highly demanding industrial applications, given its high resistance to corrosion in acidic environments and its biocompatibility. It has been recommended by the American Society for Testing and Materials (ASTM) for the manufacture of implants and by the Food Drug Administration (FDA) for the manufacture of coronary stents.

The aims of this study were the analysis and characterization of the effects of EP process factors—interelectrode gap, initial surface roughness, electrolyte temperature, current density, and EP time—on aspects of the EP process that have not been assessed in the literature such as polishing rate, final surface roughness, dimensional accuracy, and electrical consumption cost. In addition, the application of optimization methodologies in order to obtain optimum individual and multi-objectives considering criteria of polishing rate, dimensional accuracy, and electrical consumption cost. For this purpose, the response surface, overlaid contour plot (OCP), and the desirability function method (DFM) were used.

## 2. Experimental Design

The rectangular prismatic workpieces for the experimental tests were 70 × 30 × 2 mm AISI 316L stainless steel (low carbon-chromium-nickel-molybdenum austenitic stainless steel: C-0.03%, Si-0.50%, Mn-1.38%, Ni-10.08%, Cr-16.93%, Mo-2.05%, N-0.05%, S-0.01%, P-0.034%, bal. Fe). The electrolyte was composed of sulphuric acid (35%) and orthophosphoric acid (45%), the most commonly used acids for stainless steel [4,34,35]. The pilot electropolishing plant used in the tests (Figure 1) consisted of five tanks (from left to right): hot air-drying, ultrasound degreasing, washing in deionised water (1), electropolishing with mechanical agitation and temperature control, and washing with deionised water (2). The first step of the EP process involved ultrasound cleaning of the workpieces in hot water with a 5 mL/L solution of tensioactive agents to remove surface residue. Thereafter, the workpieces were washed in deionised water (1) prior to proceeding to the drying tank. After removing impurities and drying, the anodes were clamped with copper clips before submerging them into the electrolyte (electropolishing tank) with mechanical agitation and temperature control and then applying the required continuous current. After electropolishing, the workpieces were washed again in deionised water (2) to remove electrolytic debris prior to drying.

The first experimental factorial design involved five factors at two levels (2^5^) with the following factors [36,37]: initial roughness $\left({Ra}_{i}^{0}\right)$, electrode gap ($D_{i}$), temperature (*T_i_*), current density (*J_i_*), and EP time (*t_i_*). The initial surface textures were grouped into two finishing ranges: 0.5 ≤ *Ra* ≤ 0.8 µm and 1.0 ≤ *Ra* ≤ 1.3 µm, obtained by silicon carbide (SiC) disc sanding with two different average grain size sandpapers P30 (642 µm), and P60 (269 µm), respectively [38]. These selected initial finishing ranges were similar to those obtained in fine finish conditions (0.5 ≤ *Ra* ≤ 0.8 µm) and coarse finish (1.0 ≤ *Ra* ≤ 1.3 µm) in turning and milling processes. In order to examine the effects of the distance between electrodes, two electrode gaps were analyzed (Figure 1c) corresponding to the maximum and minimum cathode (-) distance in the electropolishing tank, 300 mm (*D*_1_), and 150 mm (*D*_2_), respectively, with the anode (+) always positioned in the center of the tank. The levels for the remaining factors are shown in Table 1, which includes the most widely used minimum and maximum ranges for the electropolishing of AISI 316L stainless steel.

The polishing rate, named *∆Ra*, was determined by the relative percentage decrease of the *Ra* parameter obtained after the EP process (Equation (1)), where ${Ra}_{i}^{0}$ was the initial surface roughness value of the workpiece before the EP process, and ${Ra}_{i}^{f}$ the final roughness value after the EP process. This enabled the precise determination of the relative polishing rate *ΔRa* for each workpiece, which was associated with process performance.
(1)$$\Delta Ra=\frac{Ra_{i}^{0}-Ra_{i}^{f}}{Ra_{i}^{0}}\times 100$$

The next step involved an in-depth characterization of the effects of significant EP process control parameters. A second factorial design was performed with three quantitative factors (temperature, current density, and EP time) at four levels and a qualitative factor (initial texture) at two levels, with a total of 4^3^ × 2 tests [36,37]. The initial texture was evaluated at two finishing ranges: i.e., a fine finish range $Ra_{1}^{0}$ (0.5 ≤ *Ra* ≤ 0.8 µm) and a coarse finish range $Ra_{2}^{0}$ (1.0 ≤ *Ra* ≤ 1.3 µm), which are two of the most common surface finishes processed by electropolishing. The remaining quantitative factors were analyzed at four levels, given that they could be modulated by any value within the established maximum and minimum values of the electropolishing cell. The levels analyzed in the second experiment are shown in Table 2.

Two 0.8 mm in diameter TESA GTL21 inductive probes and a measurement bolt linear travel of 4.3 mm were used to measure the specimen thickness. The measuring range probes were ±2 mm with a repeatability limit of 0.01 µm and a maximum permissible error for linearity deviation of 0.2 + 2.4 *L* µm (where *L* was in meters). Both inductive probes were placed in the TESA UPC Gauge Block Comparator (Figure 2a) for comparative measurement with nominal lengths ranging from 0.5 to 100 mm. The probe configuration consisted of two axial probes aligned opposite one another (Figure 2b), with a measuring configuration of the two probes connected in sum measurement (function +A+B) through the Digital Measuring System, with mechanical contact between measuring ball and specimen measuring face (Figure 2c). Surface roughness was evaluated with a Hommel Tester T500 class 1 contact profilometer [39], and the characterization of surface finish with the parameter roughness arithmetic mean *Ra* [40], which is the parameter most widely used in industry for evaluating surface finish [41]. Surface roughness was measured in three equidistant profiles (Figure 2e) perpendicular to the groove direction using a digital gaussian filter *λc* = 0.8 mm and a sampling length of 4 mm [42].

## 3. Results and Discussion

### 3.1. Identification of Significant Factors in the Electropolishing Process

In order to identify the significant factors in the electropolishing process, the response variable polishing rate (*ΔRa*), which measured the polishing percentage and performance of the EP process, was analyzed. For this purpose, multifactorial analysis of variance (ANOVA) was performed using a 2^5^-factorial design with the five factors under study—initial surface roughness $(Ra_{i}^{0}),$ electrode gap ($D_{i}$), electrolyte temperature (*T_i_*), current density (*J_i_*), and EP time (*t_i_*)—at two levels (high-low) [36], the results of which are shown in Table 3. The simple main effect and the interaction between two independent variables [36] of the EP process were analyzed. In the ANOVA analysis (Table 3), a 95% confidence level was applied, which implies that *p*-values for main effects or interactions greater than 0.05 were not statistically significant [36]. As shown in Table 3, the electrode gap (*D_i_*) was the only factor having no significant effect on the output variable *ΔRa*, neither in the main effect nor in the interaction effects, with all obtaining a *p*-value > 0.05. For the initial roughness $Ra_{i}^{0}$, significant effects were found in the main factor and in the interactions with temperature (B×C) and current density (B×D). Similarly, for temperature *T_i_*, significant effects were observed in the main factor and in the interaction with current density (C×D). The process control parameters of current density (*J_i_*) and EP time (*t*_i_) showed the greatest effects on the output variable *ΔRa* as indicated by the Type III Sum of Squares. Therefore, for the characterization of factor effect analysis of the EP process, only the factors that were significant in the ANOVA analysis (Table 3) were considered. In the next sections, the effects of the significant variables on different aspects of the EP process are analyzed in depth.

### 3.2. Analysis of the Electropolishing Parameter Effects on the Polishing Rate

For the analysis of the effects of significant factors on the polishing rate (*ΔRa*), a 4^3^ × 2 factorial design was performed with three quantitative factors at four levels and one qualitative factor at two levels (Table 2). The experimental data were adjusted using multivariable polynomial regression by the forward stepwise selection method, with *p*-values to simplify the models. As the models varied in the number of variables, the models were selected according to the parameter adjusted determination coefficient $R_{adj}^{2}$ (adjusted R-squared), with only the highest adjusted R-squared being selected, and prioritizing models with the least number of variables (Principle of Parsimony), and the unbiased models with Mallow’s coefficient of $C_{p}=p,$ where p=n+1 with *p* being the number of model parameters that was equivalent to the number of variables *n* plus the constant. All of the regression models obtained in this paper were diagnosed using residual analysis: all models exhibited normality in the distribution of residuals, and no significant tendencies in the residual versus fitted value plot or residual versus observation order plot were observed in any of the models analyzed [36]. The response surface method (RSM) [43] was used for the characterization of the behavior and individual optimization of significant factors on the response variable. As initial texture was a qualitative parameter, it was analyzed separately to obtain an independent regression model for each initial texture.

#### 3.2.1. Initial Texture $Ra_{1}^{0}$

Equation (2) shows the best model obtained for the initial texture $Ra_{1}^{0}$, where simple main effects and first and second-order interactions were analyzed. The model achieved a fit to the data *R*^2^ of 86.00% and adjusted *R*^2^ of 84.25%. Table 4 shows the ANOVA analysis and the *p*-values for the simple main effect and the interactions that were significant. A substantial interaction of the current density with the EP time (*J*×*t*) and with the electrolyte temperature (*T*×*J*) was observed, as evidenced by the SS Type III in Table 4.
(2)$$\Delta Ra=1.88+1.67J+3.39t-0.015T^{2}-0.02J^{2}-0.04t^{2}+0.02TJ-0.02Jt$$

Figure 3 shows the response surfaces for the four temperatures analyzed. The blue areas represent low polishing rates, green-yellow medium polishing rates, and yellow-red areas have high polishing rates. In all of the cases, a significant interaction was observed between current density and EP time, and maximum polishing rates were obtained with high current and time combinations. The initial increase in current density produced rapid improvements in process efficiency (*ΔRa*) up to *J* of ~29 A/dm^2^, but above this value, process improvement was slight with a gradually rising slope of the curve that reached maximum efficiency at *J* of ~48 A/dm^2^, after which process efficiency begins to decline. The maximum polishing ranges were obtained with current densities from 48 to 60 A/dm^2^ in all of the temperatures analyzed. A similar behavior was observed with increased EP time raising process performance *ΔRa* up to ~25 min. Above this value, performance remained constant. The maximum polishing values were obtained at 35 min in all of the temperatures analyzed. The maximum absolute polishing rate was 90.72% with a 48 A/dm^2^, 35-min, and 45 °C combination (Table 5). The differences obtained between the maximum *ΔRa* in the 35 °C to 55 °C temperature range were negligible, with a variation of 1.4% (Table 4). In contrast, there was a decline in performance at a temperature of 65 °C, with a worsening of *ΔRa* close to ~5%. Thus, temperatures above 55 °C significantly worsened process performance. A further crucial factor that should be borne in mind is that the maximum polishing rates (Table 5) required increasing current density according to the temperature of the electrolyte. Though similar maximum polishing rates were obtained in the 35 °C to 55 °C temperature range, current consumption increased from 48 to 56 A/dm^2^, respectively, which translated into higher electrical consumption costs. This may have been due to a decline in the electrical conductivity of the electrolyte, which may have been affected by the increase in electrolyte temperature.

#### 3.2.2. Initial Texture $Ra_{2}^{0}$

Equation (3) shows the best model obtained for the initial texture $Ra_{2}^{0}$, where simple main effects and first and second-order interactions were analyzed. The model achieved a fit to the data *R*^2^ of 74.52% and adjusted *R*^2^ of 72.29%. Table 6 shows the ANOVA analysis and the *p*-values for the simple main effect and the interaction that were significant. In this case, electrolyte temperature was not significant, but once again, a substantial interaction between current density and the EP time (*J*×*t*) was found in the SS Type III in Table 6.
(3)$$\Delta Ra=-22.14+2.28J+3.42t-0.015J^{2}-0.040t^{2}-0.022Jt$$

As shown in Figure 4, the response surface obtained in the model Equation (3) exhibited similar behavior to that obtained for initial texture $Ra_{1}^{0}$, except for the temperature factor. Once again, a significant interaction was observed between current density and EP time, with the highest polishing rates obtained at the highest current density and EP time combinations. Increasing current density significantly improved process performance *ΔRa* up to a current of ~29 A/dm^2^. Above this value, improvement in process performance slowed up to a current density of ~48 A/dm^2^ and then leveled off before declining. The increase in EP time exhibited similar behavior, with a rapid increase in *ΔRa* up to ~25 min, before stabilizing with no significant decline in process performance. The maximum absolute polish rate was 87.86%, obtained at 60 A/dm^2^ and 33 min. Though the differences in maximum *ΔRa* values between $Ra_{1}^{0}$ and $Ra_{2}^{0}$ textures were small (2.86%), high initial roughness textures obtained lower *Ra* roughness reduction percentages. This indicated that with higher initial roughness values, worse EP process performances were obtained, with lower polishing rates.

### 3.3. Analysis of the Electropolishing Factors Effects on Final Surface Roughness

Though the final roughness $Ra^{f}$ obtained in the EP process was an output variable correlated with processing performance *∆Ra*, it was worth analyzing the minimum *Ra* roughness values obtained in the EP process with two different initial textures $(Ra_{1}^{0}$ and $Ra_{2}^{0}),$ and their effects on final roughness under specific EP conditions.

#### 3.3.1. Initial Texture $Ra_{1}^{0}$

Equation (4) shows the best model obtained for the initial texture $Ra_{1}^{0}$, where simple main effects and first and second-order interactions were analyzed. The model achieved a fit to the data *R*^2^ of 82.73% and an adjusted *R*^2^ of 80.16%. Table 7 shows the ANOVA analysis and the *p*-values for the simple main effects and the interactions that were significant. In this case, all factors were significant, with a significant interaction between current density and the electrolyte temperature (*T*×*J*), and current density and EP time (*J*×*t*), as shown in the SS Type III in Table 7.

Figure 5 illustrates the response surfaces for the four temperatures analyzed. The blue areas represent low roughness, green-yellow medium roughness, and yellow-red areas have high roughness. A significant interaction between current density and EP time was observed, with minimum surface roughness *Ra* values being obtained for high-value combinations of *J* and *t*. For all test temperatures, the surface roughness of *Ra* declined sharply up to a current density of ~29 A/dm^2^. Above this value, the increase in current density and the improvement in surface roughness stabilized and subsequently worsened. This may be attributed to high current density values excessively increasing the material removal rate, which worsened surface roughness. The EP time had a quasi-lineal behavior in all of the temperatures, where an increase in EP time progressively reduced roughness *Ra*, with minimums obtained at 35 min for all the temperatures (Table 8). The increased temperature worsened the surface roughness *Ra* results, requiring higher current density and EP time to obtain similar roughness *Ra* results. This may be due to the diminishing electrical conductivity of the electrolyte with increased temperature. The $Ra^{f}$ absolute minimum of 0.035 μm was obtained for 46 A/dm^2^ and 35 °C (Table 8). The other minimum (Table 8) showed a worse surface finish, and the increase in electrolyte temperature required higher current density.
(4)$$Ra^{f}=0.59-0.011J-0.017t+8.9\cdot 10^{-5}T^{2}+1.67\cdot 10^{-4}J^{2}+1.89\cdot 10^{-4}t^{2}-1.65\cdot 10^{-4}TJ+1.05\cdot 10^{-4}Jt$$

#### 3.3.2. Initial Surface Texture $Ra_{2}^{0}$

Equation (5) shows the best model obtained for final roughness $Ra^{f}$ in the initial surface texture $Ra_{2}^{0}$, where simple main effects and first and second-order interactions were analyzed. The model provided a fit to the data *R*^2^ of 71.76% and adjusted *R*^2^ of 69.33%. Table 9 shows the ANOVA analysis and the *p*-values for the simple main effects and the interactions that were significant. In this case, similar to the polishing ratio results for texture initial $Ra_{2}^{0}$, the electrolyte temperature was not significant either in terms of main effects or interactions. The current density and EP time were significant, with a significant interaction between both (*J*×*t*) as shown in the SS Type III (Table 9).

Figure 6 shows the response surface obtained for this model (Equation (5)). The behavior of roughness ${Ra}^{f}$ in relation to current density and EP time was quite similar to that observed in the initial texture $Ra_{1}^{0},$ except that the EP time, in this case, did not exhibit linear behavior and slightly stabilized before worsening at high EP time. The minimum roughness of 0.147 µm was obtained for 62 A/dm^2^ at 35 min, a value 4.2 times higher than the absolute minimum obtained for the initial texture $Ra_{1}^{0}$. The most notable finding of this study was the significant effects of initial surface texture on the final roughness obtained in the EP process. For an initial roughness $Ra_{1}^{0}$ of 0.5 ÷ 0.8 µm (fine finish), the final roughness $Ra^{f}$ was ~0.045(±0.007) µm; but for initial roughness $Ra_{2}^{0}$ of 1.0÷1.3 μm (coarse finish), the final roughness ${Ra}^{f}$ was ~3.27 times higher (0.147 μm). The initial roughness ratio $\left(Ra_{1}^{0}:Ra_{2}^{0}\right)$ was ~1:1.77 versus the final roughness ratio $\left(Ra_{1}^{f}:Ra_{2}^{f}\right)$, which was ~1:3.27.
(5)$$Ra^{f}=1.31-0.023J-0.037t+1.53\cdot 10^{-4}J^{2}+4.56\cdot 10^{-4}t^{2}+2.45\cdot 10^{-4}Jt$$

### 3.4. Analysis of the Effects of Electropolishing Factors on Dimensional Accuracy

In EP processes, dimensional accuracy is a critical aspect of electropolished components that must comply with dimensional and geometric tolerances for assembly operations. Procedures for reducing the roughness of a specific surface area often overlook that material loss occurs on all of the surfaces coming into contact with the electrolyte, which significantly changes the dimensional measurements of the end product. Thus, the dimensional accuracy *∆h* was determined by measuring the thickness variations of the workpieces with respect to the initial thickness.

#### 3.4.1. Initial Texture $Ra_{1}^{0}$

Equation (6) shows the best model obtained for the dependent variable *∆h* in the initial texture $Ra_{1}^{0}$, where the simple main effects and first and second-order interactions were analyzed. The model obtained a fit to the data *R*^2^ of 89.87% and an adjusted *R*^2^ of 89.18%. Table 10 shows the ANOVA analysis and the *p*-values for the simple main effects and the interactions that were significant. All factors were significant but with a substantial interaction between the current density and the EP time (*J*×*t*), as shown in the SS Type III in Table 10.
(6)$$\Delta h=-0.081+4.99\cdot 10^{-3}T-5.12\cdot 10^{-3}J+6.30\cdot 10^{-5}J^{2}+3.71\cdot 10^{-4}Jt$$

Figure 7 shows the response surface obtained for this model (Equation (6)). The blue areas represent high dimensional accuracy, green-yellow medium dimensional accuracy, and yellow-red areas have low dimensional accuracy. As shown in Figure 7, the response surfaces obtained with model Equation (6) for the four temperatures analyzed revealed a substantial interaction between current density and EP time. The influence of the current density parameter on dimensional accuracy depended on the value (low or high) of the EP time and vice versa. At low current density values, EP time produced negligible effects on the *∆h* variable, but at high current density values, the effect of EP time produced large variations in the variable *∆h* (Figure 7). Similarly, for low EP time values, the current density also had a minor effect on the *Δh* variable, but at high EP time values, the effect of current density produced large variations in the variable *∆h* (Figure 7). The maximum *∆h* was obtained for maximum values of current density and EP time. The minimum *∆h* was achieved with a current density of 32 A/dm^2^ and a minimum 3-min EP time for all analyzed temperatures (Table 11). This was due to the 3-min current density curve reaching a minimum at 32 A/dm^2^ (Table 11). The maximum *Δh* was obtained with maximum current density (67 A/dm^2^) and EP time (36 min), with a quasi-lineal behavior of both EP parameters in relation to the variable *Δh*. As shown in Figure 7, and Table 11, dimensional variation was found to vary significantly, ranging from a minimum thickness variation of 0.031 mm to a maximum of 1.036 mm, depending on the electropolishing conditions, which underscored the significant changes in the dimensional accuracy of a product when the EP process was not optimized. As for temperature, the lowest dimensional variation was obtained at a temperature of 35 °C in all of the current density and EP time combinations. This temperature also exhibited excellent behavior in process performance *∆Ra* and surface roughness $Ra^{f}.$ Moreover, this temperature involved the lowest cost in terms of heating the electrolyte.

#### 3.4.2. Initial Texture $Ra_{2}^{0}$

Equation (7) shows the best model obtained for the response variable dimensional accuracy *Δh* for the initial texture $Ra_{2}^{0}$, where simple main effects and first and second-order interactions were analyzed. The model provided a fit to the data *R*^2^ of 85.27% and adjusted *R*^2^ of 84.53%. Table 12 shows the ANOVA analysis and the *p*-values for the simple main effects and the interactions that were significant. Similar to the initial texture $Ra_{1}^{0},$ all factors were significant, with a substantial interaction between current density and the EP time (*J*×*t*), as shown in the SS Type III in Table 12.
(7)$$\Delta h=0.097-5.10\cdot 10^{-5}J^{2}+8.30\cdot 10^{-5}TJ+3.47\cdot 10^{-4}Jt$$

Figure 8 shows the response surfaces for the model in Equation (7) for the four temperatures analyzed, which exhibited similar behavior to that obtained for initial texture $Ra_{1}^{0}$, with comparable minimum and maximum values (Table 13). Similar behavior was observed with the parameters’ current density and EP time for the initial texture $Ra_{2}^{0}.$ However, significant differences were found in *Δh_min_* values between both initial textures, particularly at a temperature of 35 °C, where *Δh_min_* was 4.2 lower in the initial texture $Ra_{1}^{0}$ (Table 11 and Table 13). These differences gradually diminished with increased temperature, with the best results being obtained at 65 °C for initial texture $Ra_{2}^{0}$. Moreover, a substantial interaction of initial texture on the variable *Δh* was observed when the material variation was at a minimum. The absolute minimum was obtained for the initial texture $Ra_{1}^{0}$ with a *Δh_min_* value of 0.031 mm for the 32 A/dm^2^, 3-min, 35 °C combination, which was far below the other minimums obtained. In relation to *Δh_max_*, the values were very similar for both textures (Table 11 and Table 13), with no significant differences. This indicated dimensional variation *Δh* was not dependent on initial texture when there was an excessive material loss.

### 3.5. Analysis of the Effects of Electropolishing Factors on Electrical Consumption Cost

The total cost of an electrochemical polishing operation must include all of the individual costs involved. Certain costs are fixed and independent of the product to be electropolished, such as labor costs, the price of the electrolyte, and the energy consumed to heat the electrolyte, among others. Thus, the present study was restricted to the analysis of the electrical consumption cost of the electrical current consumed in the electropolishing operation without taking into account other additional costs. For this analysis, the parameter specific electrical energy consumption *C_e_* was used, which was defined as the kilowatts x hour consumed in the electropolishing of a surface per square decimetre (kWh/dm^2^). This parameter, which did not depend on the size of the treated surface, served to estimate the cost per unit of an electropolished surface. In order to determine the *C_e_* parameter, it was necessary to determine the current curve of the electropolishing cell used in the experimental trials (Figure 9): that is, the relation between the current intensity applied to the workpiece and the difference in potential obtained between the electrodes.

The voltage-intensity curve obtained in the electropolishing cell was defined by Equation (8):(8)U=2.24+0.27I

The electrical energy consumption *C* in an EP operation was defined as the electrical power *P* applied to the circuit during the period of time *t* of the process, as determined by Equation (9):(9)C=Pt

In a continuous current (CC) circuit, the electrical power *P* consumed during a certain period of time was defined as the product of the potential difference *U* between the electrodes due to current intensity *I* circulating through the circuit (Equation (10)):(10)P=UI

On the basis of Equations (9) and (10), the consumption of electrical energy *C* of the EP process was defined as the product of the potential difference between the electrodes due to the current intensity applied to the CC circuit and the electrical current application time according to Equation (11):(11)C=UIt

Thus, the specific electrical energy consumption *C_e_* in the EP process, defined as the electrical energy *C* consumed per unit of electropolished surface, is defined by Equation (12):(12)$$C_{e}=\frac{C}{A}=\frac{UIt}{A}$$
where *A* was the total electropolished surface. As the relationship between the potential difference *U* and current intensity *I* in the workpieces was defined by Equation (8), the specific electrical energy consumption *C_e_* was defined by Equation (13):(13)$$C_{e}=\frac{(2.24+0.27I)It}{A}=\frac{2.24It+0.27I^{2}t}{A}=\frac{2.24It}{A}+\frac{0.27I^{2}t}{A}=\frac{2.24It}{A}+\frac{0.27AI^{2}t}{A^{2}}$$

When in Equation (13), the relation *I/A* corresponding to current density *J* (Equation (14)) applied to the EP process was substituted, specific electrical energy consumption *C_e_* was defined according to current density *J* and EP time *t* (Equation (15)), which were control parameters commonly used in electropolishing processes.
(14)$$J=\frac{I}{A}$$
(15)$$C_{e}=2.24Jt+0.27AJ^{2}t$$

When in Equation (15), parameter *A* was replaced by the numerical value of the electropolished workpiece surface (0.39 dm^2^), the specific electrical energy consumption *C_e_* for the electropolishing of the experimental workpieces was defined in Equation (16):(16)$$C_{e}=2.24Jt+0.1053J^{2}t$$

Figure 10 shows the response surface for specific electrical energy consumption *C_e_* of the workpieces according to Equation (16). The blue areas represent a low cost, the green-yellow medium cost, and the yellow-red areas are a high cost. A substantial interaction was found between current density and EP time, which was quite similar to that obtained for the *Δh* variable. At low current density values (10 A/dm^2^), the effects of EP time on the cost *C_e_* were negligible, with values of 0.002 and 0.019 kWh/dm^2^ for combinations of 3 min-10 A/dm^2^ and 36 min-10 A/dm^2^, respectively. However, at high current density values of (67 A/dm^2^), the EP time effect produced a large variation in the cost *C_e_* with values of 0.031 and 0.373 kWh/dm^2^ for the combinations 3 min-67 A/dm^2^ and 36 min-67 A/dm^2^, respectively. Similarly, for the EP time parameter, the influence on the cost varied according to the (low or high) current density value. The maximum value of the cost *C_e_* was 0.373 kWh/dm^2^ for 67 A/dm^2^ and 36 min. Though both parameters exhibited quasi-lineal behavior, the slope increased from the medium values of *J* and *t*.

### 3.6. Multi-Objective Optimization by Overlaid Contour Plot

Normally, the optimization of the electropolishing processes is fundamentally based on minimizing the surface roughness value or on establishing a specific range of surface finish without taking into account dimensional variations or the cost of treating a product. However, both aspects are critical in the electropolishing of functional products. Thus, this study analyzed the simultaneous optimization of these three variables using the overlaid contour plot (OCP) method. This method establishes optimum areas in terms of multiple application criteria. The simultaneous optimization of these three variables was established in terms of process performance percentage ranges in order to establish the optimum maximum roughness, the loss of dimensional accuracy, and the minimum cost of the EP operation for each polishing range. Moreover, a point of equilibrium between the three optimum points was established, which enabled the multi-objective optimization of each polishing range.

The contour graphs were obtained using the multivariable polynomial regression models for the response variables *ΔRa*, *Δh,* and *C_e_* from the previous sections (Equations (2), (3), (6), (7), and (16)). The polishing ranges *ΔRa* were established at 10% intervals, and from a productive perspective, only polishing ranges above 60% were analyzed. For the variable *Δh*, contour lines were established every 0.1 mm, and for the variable *C_e_*, every 0.04 kWh/dm^2^. Taking into account these aspects, Figure 11 and Figure 12 show the OCP method for the three variables under analysis. For each polishing range, the maximum polishing point *P^i^*, minimum dimensional variation *D^j^*, and minimum cost *C^i^* were determined, where *i* was the polishing rate. Finally, the optimum multi-objective point *O^i^* was graphically determined by minimizing the maximum distance between these three optimum points.

In Figure 11, the optimum areas for the polishing ranges of 60, 70, 80, and 90% for initial texture $Ra_{1}^{0}$ were highlighted with a white background. The 90% optimum polishing area was defined by a minimum reduction in *ΔRa* of 90%, a maximum variation in dimensional accuracy *Δh* of 0.7 mm, and maximum consumption *C_e_* of 0.24 kWh/dm^2^. In this optimum area, maximum polishing *P*^90^ was 90.50%, obtained at 52 A/dm^2^ and 33 min, with a dimensional variation of 0.680 mm and a cost of 0.24 kWh/dm^2^ (Table 14). The minimum dimensional variation point *D*^90^ of 0.620 mm was obtained at 50 A/dm^2^ and 31 min, with a consumption of 0.210 kWh/dm^2^ and a polishing rate of 90.0%. The minimum cost point *C*^90^ coincided with the point *D*^90^ as the optimum area was very small. Finally, the multi-objective point *O*^90^ corresponded with a polishing rate of 90.37% obtained at 51 A/dm^2^ and 32 min, with a dimensional variation of 0.650 mm, and a consumption of 0.230 kWh/dm^2^. Thus, the individual and multi-objective optimum were obtained in the polishing range of *ΔRa* ≥ 90%, which allow to establish different strategies of optimum EP process conditions depending on the customer’s requirement.

As the 90% polishing range was only obtained for the initial texture $Ra_{1}^{0}$, the 80% range was used for the comparison between initial textures. The 80% polishing area (Figure 11 and Table 14) was limited to an 80% minimum reduction in *Ra*, with a dimensional variation and electrical energy consumption below 0.4 mm and 0.12 kWh/dm^2^, respectively. In this range, the maximum polishing *P*^80^ of 82.06% was obtained at 44 A/dm^2^ and 21 min, with a *Δh* of 0.380 mm and a *C_e_* of 0.120 kWh/dm^2^. The minimum variation in dimensional accuracy *D*^80^ was 0.350 mm at 44 A/dm^2^ and 19 min, with 80.00% of *ΔRa*, and 0.100 kWh/dm^2^ of *C_e_*. The point of minimum consumption *C*^80^ was 0.100 kWh/dm^2^ at 40 A/dm^2^ and 21 min, with 80.00% of *ΔRa*, and 0.350 mm in *Δh*. For the multi-objective point *O*^80^, a *ΔRa* of 80% was obtained at 42 A/dm^2^ and 20 min, with a *Δh* of 0.350 mm and *C_e_* of 0.100 kWh/dm^2^. In this case, there were greater differences that improved individual and multi-objective optimization. Table 14 shows the optimum points obtained for the other polishing ranges analyzed of 70% and 60%.

Figure 12 depicts the contour surface graphs of the initial texture $Ra_{2}^{0}$ where the maximum absolute polishing *P_max_* range of 87.86% was observed for the electropolishing conditions of 60 A/dm^2^ and 33 min, with a variation in dimensional accuracy of 0.720 mm and a consumption of 0.26 kWh/dm^2^. Similar to the results of the individual analyses, the initial texture $Ra_{2}^{0}$ exhibited the worst EP process performance with lower polishing rates than the initial texture $Ra_{1}^{0}.$
Figure 11 and Figure 12 show that in the initial texture $Ra_{2}^{0}$, the optimization areas required higher current density and EP time to obtain similar polishing rates than initial texture $Ra_{1}^{0}$.

The 80% polishing area was delimited by a minimum reduction in *Ra* of 80% and a maximum *Δh,* and *C_e_* of 0.6 mm and 0.16 kWh/dm^2^, respectively (Figure 12). The maximum polishing *P*^80^ was 81.50% for the 51.5 A/dm^2^ and 21-min polishing condition, which was very similar to that obtained for the initial texture $Ra_{1}^{0}$ (82.06%), but with a higher current density of 44 A/dm^2^. The optimum dimensional accuracy *D*^80^ was 0.531 mm at 51.5 A/dm^2^ and 21 min, which was worse than that obtained for the initial texture $Ra_{1}^{0}$ (0.350 mm). The optimum consumption *C*^80^ of 0.144 kWh/dm^2^ was obtained at a current density of 44 A/dm^2^ and 25 min, and worse than that obtained for the initial texture $Ra_{1}^{0}$ (0.100 kWh/dm^2^). As for the optimum multi-objective *O*^80^, the 80.47% polishing rate was obtained at 47.75 A/dm^2^ and 23 min, with a dimensional variation of 0.539 mm and a cost of 0.150 kWh/dm^2^. As shown in Figure 12, all of the optimum points obtained for the initial texture $Ra_{2}^{0}$ were worse than in the initial texture $Ra_{1}^{0}$, with worse dimensional accuracy and electrical consumption due to the higher current density. Table 15 shows the optimum points for all of the areas analyzed.

### 3.7. Multi-Objective Optimization Based on the Desirability Function

#### 3.7.1. Fundamentals of the Desirability Function Method

The desirability function method (DFM) was used for multi-objective optimization as it is one of the most extensively used methods for the optimization of multiple responses in productive processes. This method was initially proposed by Harrington [44] in 1965. In the present study, the discontinuous functions of Derringer and Suich [45] were used, as defined by Equations (12)–(14). The application of the method required determining *m* response functions *y_i_* for simultaneous optimization. The *d_i_*(*y_i_*) denominated the individual desirability function of each response *y_i_*, where *i = 1…m*. For each response *y_i_,* the individual desirability function *d_i_*(*y_i_*) was determined that expressed the desirability of a response value on a desirability scale from 0 to 1 (0 ≤ *d_i_*(*y_i_*) ≤ 1), where the value 0 was associated with a completely undesirable response, and the value 1 a completely desirable response. Each *y_i_* response appeared in one of three forms described in sections a, b, or c, depending on whether the response was maximized, minimized, or obtained a specific objective value. If the response fully accomplished the proposed objective, the desirability function had a value *d_i_*(*y_i_*) = 1 if the response reached the total undesirability value *d_i_*(*y_i_*) = 0. The first step to designing the global and composite desirability function *D* was to separately build the individual functions *d_i_*(*y_i_*) for each response *y_i_* as follows:
(a)If the objective was to maximize the response *y_i_*, the desirability function *d_i_*(*y_i_*) was defined by Equation (17), where *L* was the minimum objective of the response, and *H* was the maximum objective:(17)$$d_{i}^{max}(y_{i})=\{\begin{array}{c} 0\;\;\;\;\;\;\;\;if\;\;\;\;y_{i}<L\\ {\left(\frac{y_{i}-L}{H-L}\right)}^{s}\;if\;\;\;\;L\leq y_{i}\leq H\\ 1\;\;\;\;\;\;\;\;if\;\;\;\;y_{i}>H \end{array}$$
when the objective *H* of the response *y_i_* was its maximum value, then *d_i_*(*y_i_*) = 1; and if the objective *L* of the response *y_i_* was its minimum value, then *d_i_*(*y_i_*) = 0.(b)If the objective was to minimize the response *y_i_*, the desirability function *d_i_*(*y_i_*) was defined by Equation (18):(18)$$d_{i}^{min}(y_{i})=\{\begin{array}{c} 1\;\;\;\;\;\;\;\;if\;\;\;\;y_{i}<L\\ {\left(\frac{y_{i}-H}{L-H}\right)}^{s}\;if\;\;\;\;L\leq y_{i}\leq H\\ 0\;\;\;\;\;\;\;\;if\;\;\;\;y_{i}>H \end{array}$$When the objective *L* of the response *y_i_* was its minimum value, then *d_i_*(*y_i_*) =1; and if the objective *H* of the response *y_i_* was its maximum value, then *d_i_*(*y_i_*) = 0.(c)If the response variable was to be maintained at the objective value *T*, the desirability function was defined by Equation (19):(19)$$d_{i}^{target}(y_{i})=\{\begin{array}{c} 0\;\;\;\;\;\;\;if\;\;\;\;y_{i}<L\\ \begin{array}{c} {\left(\frac{y_{i}-L}{T-L}\right)}^{s}\;if\;\;\;\;L\leq y_{i}\leq T\\ {\left(\frac{y_{i}-H}{T-H}\right)}^{t}\;if\;T\leq y_{i}\leq H \end{array}\\ 0\;\;\;\;\;\;\;\;if\;\;\;\;y_{i}>H \end{array}$$

The parameters *s* and *t* corresponded to the weight factor of desirability function *d_i_*(*y_i_*) of the response *y_i_* and determined the shape of the function *d_i_*(*y_i_*) between the 0 and 1 points (Figure 13). For *s* >1 and *t* > 1, the function was concave, implying the desirability individual value was quite low unless the response was close to reaching its objective value. This indicated that the higher the values, the greater the significance of the response value in meeting the objective. For *s* < 1 and *t* < 1, the function was convex, implying the individual desirability value was large even when the response was from its objective value. In comparison, if *s* = *t* = 1, the desirability function increased linearly from 0 to 1.

The composite desirability *D* was determined as the geometric mean of the individual desirabilities *d_i_*(*y_i_*) obtained for each response *y_i_* where *m* was the number of optimized responses (Equation (20)). The importance factor of the response enabled the weighting of the desirability function of each individual response accordingly:(20)$$D={\left[d_{1}(y_{i})d_{2}(y_{2})\ldots d_{m}(y_{m})\right]}^{1/m}$$

#### 3.7.2. Multi-Objective Optimization with the Desirability Function

In this study, three simultaneous multiple optimization criteria were assessed, i.e., the polishing rate (*ΔRa*) that defined the percentage in the reduction of surface finish, the variation in dimensional accuracy (*Δh*) that defined the reduction in workpiece thickness, and the electrical consumption cost (*C_e_*) of the electropolishing operation. It was assumed the three criteria were of equal importance, so the individual desirability of each response *d_i_*(*y_i_*) was equally weighted in order to determine global desirability (*D*) (Equation (15)). For the potentials *s* and *t*, the linear form of the desirability function was defined as *s* = *t* = 1. The desired cut-off value for each response was established according to the minimum and maximum values considered to be satisfactory for each response. Polishing rates of an EP process were considered satisfactory as from 50% or higher; hence, the cut-off value was set within the interval of 50 ≤ *ΔRa* ≤ 100%. For the variation in dimensional accuracy, values above 1 mm were considered unsatisfactory, and 0 mm was the optimum value, so the interval was defined as 0 ≤ *Δh* ≤ 1 mm. In relation to cost, the process was considered unfeasible when costs were higher than 0.30 kWh/dm^2^, with optimum consumption obtained at 0 kWh/dm^2^; hence, the interval was defined as 0 ≤ *C_e_* ≤ 0.30 kWh/dm^2^.

The configuration of the desirability model for the initial texture $Ra_{1}^{0}$ is shown in Table 16. High and low response values were obtained according to the best fit of the individual regression models generated by the method of the desirability function. The target value was determined according to the proposed objective, with the option of maximizing or minimizing the response analyzed. In this case, the aims of the responses were to maximize *ΔRa* and minimize *Δh* and *C_e_*. The weight and importance factors of the responses were equal to 1, given that the objective was to ascertain a simultaneous solution for optimizing all three responses.

Table 17 shows the five best results of the highest composite desirability results. The optimum solution was the first one, with a desirability value of 0.88 obtained at a temperature of 35 °C, a current density of 42.81 A/dm^2^, and an EP time of 5.66 min. For this EP condition, the variable output values were 62.39% for *ΔRa*, 0.066 mm for *Δh*, and 0.023 kWh/dm^2^ for *C_e_*.

Table 18 shows the adjusted values obtained for the desirability function, which were estimated points of the mean predicted response values. The adjusted polishing rate was 62.39%, the dimensional variation was 0.066 mm, and the cost was 0.023 kWh/dm^2^. In the second column, the standard error of fit (SE Fit) was obtained, which was the precision in estimating the mean predicted response. The fit error values for the three responses analyzed ranged from 0.41 to 4%. The fourth column (95% CI) indicates the confidence interval of the responses with a probability of 95% for the mean response.

Figure 14 and Figure 15 show the multi-objective optimization graph for the three responses analyzed (*ΔRa*, *Δh*, *C_e_*) using the desirability function. Each column on the graph corresponds to one of the three process control parameters (*T*, *J*, *t*), and each row corresponds to one of the three response variables (*ΔRa*, *Δh*, *C_e_*). In the top row, the maximum and minimum experimental values for each EP parameter appear in black, and the optimum maximum composite desirability value is in red. The second row contains the maximum composite desirability *D* for all three responses and the behavior curve for each EP process control parameter. The rows below show the predicted response and the individual desirability values for each output variable. The vertical red lines represent the optimum configuration obtained, relating the output variables to the EP process parameters.

For initial texture $Ra_{1}^{0}$, a composite desirability (*D*) of ~0.88 was obtained, indicating excellent multi-objective optimization. As shown in Figure 14, composite desirability depended on the three process control parameters (*T*, *J*, *t*), with all having significant effects. Both temperature and EP time exhibited a quasi-lineal behavior, with an increase in either or both parameters having a negative effect on *D*. The optimum multi-objective was obtained for the minimum temperature of 35 °C and a low EP time of ~5.67 min, which was quite close to the minimum experimental value of 3 min. Current density had a second-order polynomial behavior, with an optimum multi-objective of ~42.82 A/dm^2^, quite close to the individual optimum of 48 A/dm^2^ in Table 5. The *J* and *t* parameters had the greatest impact on the composite desirability of D, with process control parameters having the greatest negative effects on the value of *D*. The optimization based on the desirability function obtained an optimum multi-objective (Figure 12, *ΔRa* = ~62.39%, *Δh* = ~0.066 mm, and a *C_e_* = ~0.002 kWh/dm^2^) with a high polishing rate, and a significant reduction of thickness variation and cost in comparison to the optimum multi-objective obtained by OCP (Table 14, *ΔRa* = 61.34%, *Δh* = 0.141 mm, and *C_e_* = 0.036 kWh/dm^2^).

In relation to individual desirability (Figure 14), the temperature was found to be the parameter having the smallest effect on the individual desirability of the response variables, with no effect on cost *C_e_* and a slight impact on the polishing rate and dimensional variation. The minimum temperature analyzed (35 °C) obtained the maximum polishing rate and the minimum dimensional variation. The greatest impact of current density was on the desirability of the *ΔRa* response, with an optimum multi-objective of 42.82 A/dm^2^, very close to the maximum of ~48 A/dm^2^ obtained in the individual analysis. However, the effect of current density on the desirability of *Δh* and *C_e_* was small, given that an increase in the value of this parameter produced a small increase in either desirability. EP time was the parameter having the greatest impact on the desirability of the *Δh* and *C_e_* responses, with an increase in either significantly worsening the desirability of both responses. Thus, the optimum multi-objective was achieved with low EP time values of 5.66 min, which minimized dimensional variation and cost. These results confirmed those obtained in the individual analysis of the response variables in the above sections. The lowest desirability was obtained for the *ΔRa* response with a value of ~0.75. This response improved significantly with increased *J* or *t*, but both parameters combined, mainly *t*, worsened the desirability of *Δh* and *C_e_*. The individual desirability of the *Δh* and *C_e_* responses were ~0.97 and ~0.95, respectively, very close to the maximum desirability.

The configuration of the desirability model for the initial texture $Ra_{2}^{0}$ is shown in Table 19. As for initial texture $Ra_{1}^{0}$, the aims of the responses were to maximize *ΔRa* and minimize *Δh* and *C_e_*. The weighting and importance of the responses were assumed to be equal, as the objective was to determine a solution for the simultaneous and equal optimization of all three responses.

Table 20 shows the five best results with the highest composite desirability values. The optimum solution was the first option with the highest composite desirability of 0.82, the highest polishing rate of 56.45%, a *Δh* of 0.192 mm, and a *C_e_* of 0.025 kWh/dm^2^. This optimum solution was obtained with a temperature of 41.97 °C, a current density of 58.36 A/dm^2^, and a 3-min EP time. These EP conditions obtained the highest composite desirability. In all of the five solutions, the initial texture $Ra_{2}^{0}$ required higher current and temperature conditions than the initial texture $Ra_{1}^{0}$, and obtained lower polishing rate with greater dimensional variation and cost. This confirmed the results of the individual analysis.

Table 21 shows the adjusted values for the desirability function, which were estimated points of the mean predicted response values. As can be observed, the adjusted polishing rate was 56.45%, dimensional variation was 0.192 mm, and cost in 0.025 kWh/dm^2^. The adjusted error of the three responses analyzed increased in initial texture $Ra_{1}^{0}$ from 0.48 to 5.57%. The fourth column (95% CI) indicates the confidence interval of the responses with a 95% probability for the mean response.

Figure 15 exhibits a graph of the simultaneous optimization of the three responses analyzed (*ΔRa*, *Δh*, *C_e_*) using the desirability function for initial texture $Ra_{2}^{0}$. A composite desirability (*D*) of ~0.82 was obtained, indicating an excellent degree of multi-objective optimization. As can be observed, composite desirability depended fundamentally on the current density and EP time parameters, with the effect of temperature on composite desirability being almost negligible. The optimum multi-objective was obtained at a temperature of 41.97 °C, very close to the minimum experimental value (35 °C). Increased current density improved the *D* value, and the optimum was obtained at a *J* of 58.36 A/dm^2^, but with no further improvement. Increased EP time reduced the *D* value, and the optimum being the minimum value of 3 min, but with no further improvement. This optimum multi-objective obtained a *ΔRa* of 56.45%, a *Δh* of 0.192 mm, and a *C_e_* of 0.025 kWh/dm^2^, with a worse polishing rate and greater dimensional variation and cost than in initial texture $Ra_{1}^{0}$. In this case, the optimization based on the desirability function obtained an optimum multi-objective with a lower polishing rate (~4% difference) but with a significant reduction of thickness variation and cost in comparison to the optimum multi-objective obtained by OCP (Table 15, *ΔRa* = 60.82%, *Δh* = 0.289 mm, and *C_e_* = 0.061 kWh/dm^2^),

In relation to individual desirability, Figure 15 shows temperature was the parameter with the smallest effect on individual desirability of the response variables, with no effect on cost *C_e_* and a negligible impact on the polishing rate and dimensional variation. At a low temperature (41.97 °C), the maximum polishing rate and minimum dimensional variation were obtained. The maximum effect of current density on the *ΔRa* response desirability was the optimum multi-objective of 58.36 A/dm^2^, which was very close to the maximum 60 A/dm^2^ obtained in the individual analysis. However, the effect of current density on the desirability of *Δh* and *C_e_* had a minor impact. EP time was the parameter having the greatest impact on the desirability of the *Δh* and *C_e_* responses, and an increase in this parameter significantly worsened the desirability of both responses. Thus, the optimum multi-objective was achieved with a minimum EP time value (3 min), which minimized dimensional variation and cost. These results confirmed those obtained in the individual analysis of the response variables in the above sections for this initial texture. The lowest desirability was obtained for the *ΔRa* response with a value of ~0.64. This response could improve with increased EP time but would significantly worsen the desirability of *Δh* and *C_e_*. The individual desirability of the *Δh* and *C_e_* responses were ~0.90 and ~0.94, respectively, quite close to maximum desirability but with little scope for improvement.

Similar to the results obtained in the individual analysis, the application of the DFM methodology to the initial texture $Ra_{1}^{0}$ obtained higher polishing rates with less dimensional variation and cost. Though the difference in cost was small, the reduction in thickness was nearly 3 times less in initial texture $Ra_{1}^{0}$ than in initial texture $Ra_{2}^{0}$, with a 6% improvement in polishing rate. This confirmed the results of the individual analysis.

### 3.8. Discussion

An in-depth review of the literature on the electropolishing of stainless steel was performed in order to compare and discuss the results of the present paper with the findings of other studies. To our knowledge, there are no published studies on the electropolishing of stainless steel that can be compared to the present paper. As shown in the review of Łyczkowska-Widłak et al. [35], where an extensive comparison of studies on EP processes was performed, a valid comparison with the results of the present study is impossible owing to the different EP conditions, work materials, and electrolyte compositions that have been used. Nevertheless, there are a few similarities with other works that allow us to draw a comparison focusing only on partial results. In the majority of the published studies on the EP of stainless steels [4,34], the general conclusions established that the parameters current density, EP time, and electrolyte temperature were the most relevant in the EP process, in particular in relation to surface finish, which coincides with the results of the present study. Only the study published by Lin and Hu [16] presented certain similarities, but it was applied to different stainless steel (AISI 304), with different acid compositions, the addition of glycerol, and a reduced number of experimental tests. These authors identified current density, EP time, and electrolyte temperature as the most relevant parameters in the EP process, which coincides with the results obtained in this paper. The discrepancies with this study were related to the EP time interaction and the importance of current density. Liu and Hu [16] reported the marginal effect of current density and no interaction of EP time with other EP parameters. In the present study, current density was the most relevant EP parameter in the EP process, with substantial interactions with EP time in all variable responses (*ΔRa*, *Δh*, and *C_e_*) and both initial textures. Electrolyte temperature was only significant in the initial texture with lower surface roughness (0.5 ≤ *Ra* ≤ 0.8 μm), which was possibly due to the minor effect of current density and EP time on variable response. The present study examined the temperature range most commonly used in the electropolishing of stainless steels [34,35], concluding that the best results were achieved at ~35 °C, a temperature similar to the 30 °C of Lin and Hu [16]. The temperature conclusions were not possible to contrast with other studies due to the diversity of the results and experimental conditions [35].

As previously mentioned, certain results obtained in the present study could not be contrasted owing to the dearth of published studies in the literature. The electropolishing of surfaces with lower roughness (0.5 ≤ *Ra* ≤ 0.8 μm) produced improvements in polishing rate, final surface finish, dimensional accuracy, and electrical consumption cost versus textures with higher initial roughness (1.0 ≤ *Ra* ≤ 1.3 μm). In relation to the effects, dimensional accuracy, current density, and EP time were the most relevant parameters, having significant interactions with temperature. EP conditions with high current densities (~67 A/dm^2^) and EP times (36 minutes) produced thickness reductions of more than 1 mm, causing critical dimensional defects in the product. The electrical consumption cost, which did not depend on experimental data, was conditioned by current density and EP time, which were the parameters that determined the electrical power supplied to the EP process, with a higher dependence on current density (Equation (16)). The response surface methodology adequately defined the EP parameters’ effects on the individual objectives and precisely established each individual optimum objective. The optimum global objective obtained by the desirability function was better in terms of cost and dimensional accuracy. This method exhibited a limited EP process control but with the possibility of weighting the objectives. In contrast, the overlaid contour plot established the optimum areas per polishing ranges, with better control of EP process conditions in all polishing ranges. These two methodologies were efficient and provided adequate solutions for simultaneous multi-objective optimization.

## 4. Conclusions

In this study, the effects of the interelectrode gap, initial surface roughness, current density, EP time, and electrolyte temperature were analyzed in the electropolishing of AISI 316L stainless steel. Individual and multi-response optimization methodologies were applied to determine optimum electropolishing conditions on the basis of polishing rate, final surface roughness, dimensional accuracy, and electrical consumption cost criteria., The following conclusions may be drawn bearing in mind the results of this study:
Current density and EP time were the parameters having the greatest effects on the EP process, affecting polishing rate, final surface finish, dimensional accuracy, and power consumption cost. The increase in both parameters improved performance, reaching maximum *ΔRa* values of 85–91% at current density intervals of 48–60 A/dm^2^ and a 31–35 min EP time. Above these maximum values, the increase in both parameters produced a stabilization and a subsequent worsening of the EP process performance.Electrolyte temperature was the EP parameter with the least effect on the polishing rate and final roughness. It was only significant in initial texture $Ra_{1}^{0}.$ The maximum polishing rate obtained was 90.72% at 45 °C, but with variations below ~5% in the other temperatures analyzed. The 35 °C temperature was the best option owing to the excellent results in the EP process, i.e., polishing rate, final roughness, dimensional accuracy, and electrical consumption cost.The initial texture was significant on all criteria analyzed, but especially on the polishing rate and the final roughness obtained. The best polishing rates (89–91%) were obtained for initial texture $Ra_{1}^{0}$, with less dimensional variation and cost in all of the temperatures analyzed. The initial texture $Ra_{2}^{0}$ obtained a maximum polishing rate of 87.9%, but with greater dimensional variation and higher electrical consumption cost. The initial texture $Ra_{1}^{0}$ obtained the best final roughness with a minimum of ~0.03 µm in comparison to the initial texture $Ra_{2}^{0}$ of ~0.15 µm.As for dimensional accuracy (*Δh*), a highly significant interaction was found between current density and EP time. At low values, the modulation of both parameters had a minor effect on *Δh* with variations below ~0.150 mm, but at high values, more than 1 mm in thickness variation was observed. The lowest *Δh* was obtained at 35 °C in all of the cases analyzed.Electrical consumption cost (*C_e_*) also showed a significant interaction between current density and EP time. At low values, the modulation of both parameters had a negligible effect on *C_e_* with consumption at the interval of 0.002–0.031 kWh/dm^2^; but high values reached a maximum consumption of 0.373 kWh/dm^2^. The lowest cost was obtained at a 35–45 °C temperature range.OCP methodology determined the optimum polishing areas per polishing ranges, where the optimum individual and multi-objectives were determined using a graphical method under EP process performance, dimensional accuracy and cost criteria. This methodology allows better control of EP process conditions by polishing ranges. The results obtained with OCP confirmed the results of the individual RSM analysis.DFM methodology determined the optimum global multi-objective with excellent composite desirability values for both textures and enabled the weighting of the responses. The optimum multi-objective with DFM obtained better dimensional accuracy and cost than that obtained with OCP. The results obtained with DFM using independent models confirmed the results of the individual RSM analysis and the multi-response OCP analysis.

This study provides tools for decision-making for the individual and multi-objective optimization of the electropolishing process applied to AISI 316L stainless steel according to surface quality, dimensional accuracy, and electrical consumption cost criteria.

## Figures and Tables

**Figure 1 materials-16-01770-f001:**
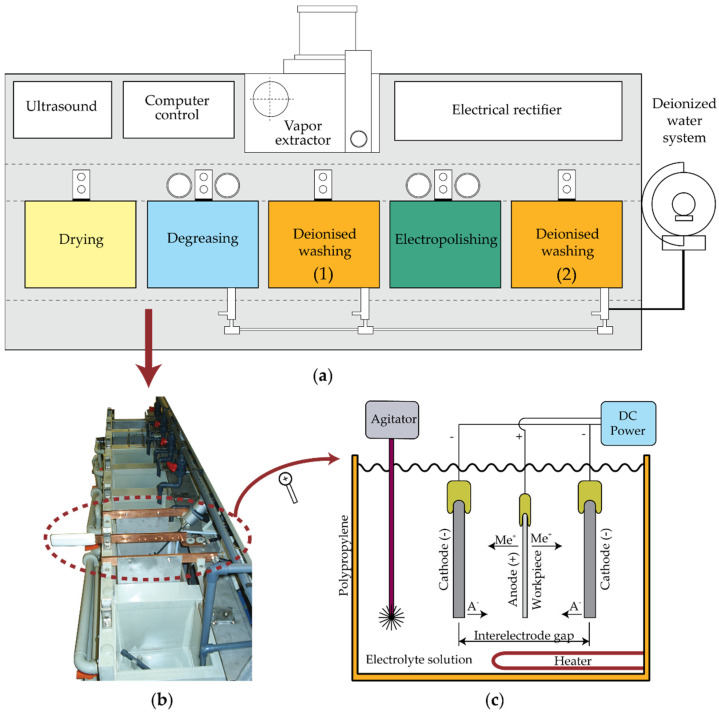
Electropolishing pilot plant: (**a**) Structure, components, and electropolishing stages, (**b**) Experimental electropolishing pilot plant; (**c**) Functional diagram of electropolishing cell.

**Figure 2 materials-16-01770-f002:**
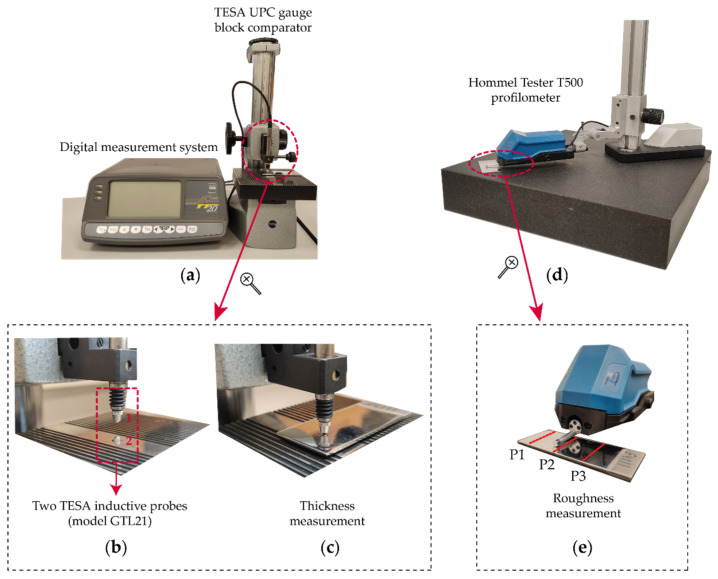
Experimental setting for the thickness and roughness measurement of the specimens: (**a**) Tesa UPC gauge block comparator; (**b**) Inductive probes aligned opposite one another for thickness measurement; (**c**) Specimen thickness measurement; (**d**) Hommel Tester T500 profilometer; (**e**) Specimen roughness measurement.

**Figure 3 materials-16-01770-f003:**
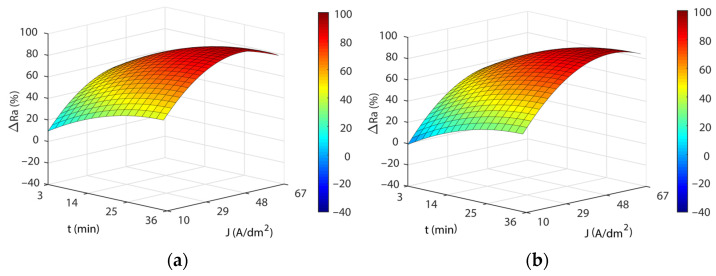
Effects of the EP process control parameters on process performance *ΔRa* for the initial texture $Ra_{1}^{0}$: (**a**) 35 °C, (**b**) 45 °C, (**c**) 55 °C, and (**d**) 65 °C.

**Figure 4 materials-16-01770-f004:**
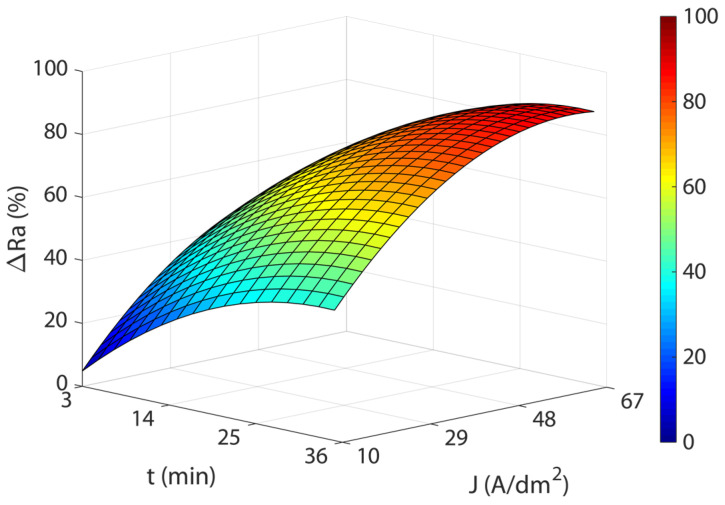
Effects of EP parameters on electropolishing rate *ΔRa* for initial texture $Ra_{2}^{0}$.

**Figure 5 materials-16-01770-f005:**
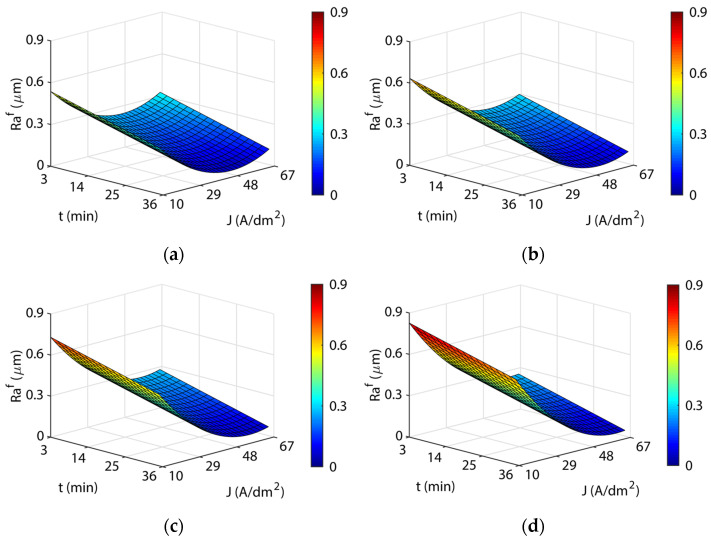
EP parameters effect on final roughness ${Ra}^{f}$ for the initial texture $Ra_{1}^{0}$: (**a**) 35 °C; (**b**) 45 °C; (**c**) 55 °C; and (**d**) 65 °C.

**Figure 6 materials-16-01770-f006:**
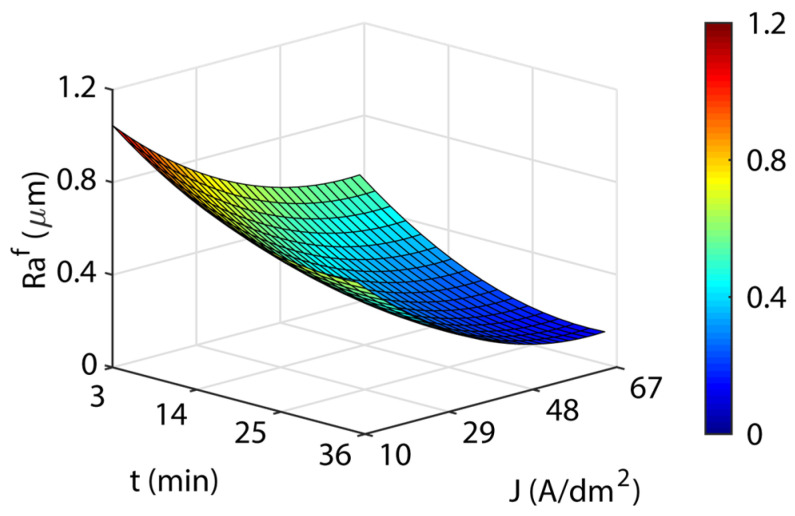
EP parameters effect on final roughness ${Ra}^{f}$ of the initial texture $Ra_{2}^{0}$.

**Figure 7 materials-16-01770-f007:**
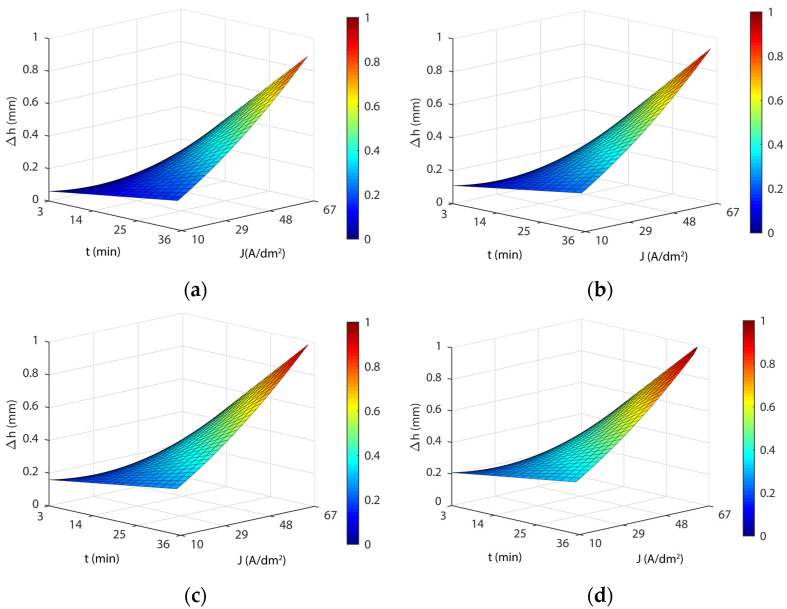
EP parameters effect on dimensional accuracy (*Δh*) for the initial texture $Ra_{1}^{0}$: (**a**) 35 °C; (**b**) 45 °C; (**c**) 55 °C; and (**d**) 65 °C.

**Figure 8 materials-16-01770-f008:**
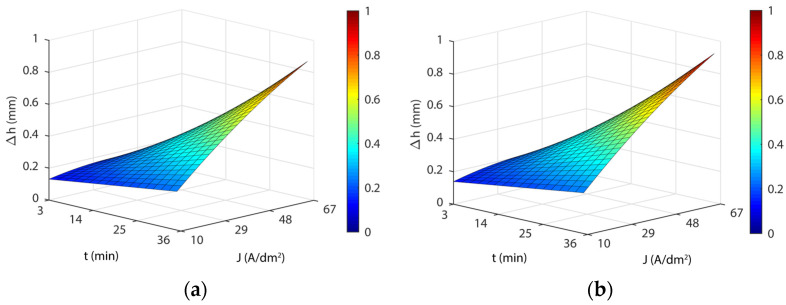
EP parameters effect on dimensional accuracy (*Δh*) for the initial texture $Ra_{2}^{0}$: (**a**) 35 °C; (**b**) 45 °C; (**c**) 55 °C; and (**d**) 65 °C.

**Figure 9 materials-16-01770-f009:**
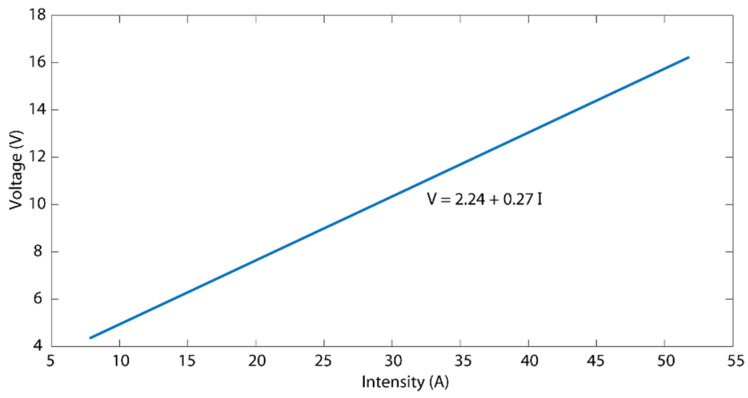
Voltage-intensity relationship in the electropolishing cell used in the experimental tests.

**Figure 10 materials-16-01770-f010:**
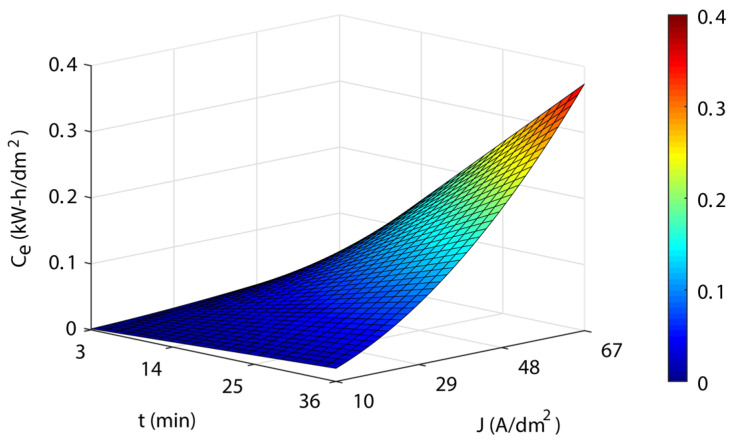
EP parameters effects on the specific electrical energy consumption cost *C_e_*.

**Figure 11 materials-16-01770-f011:**
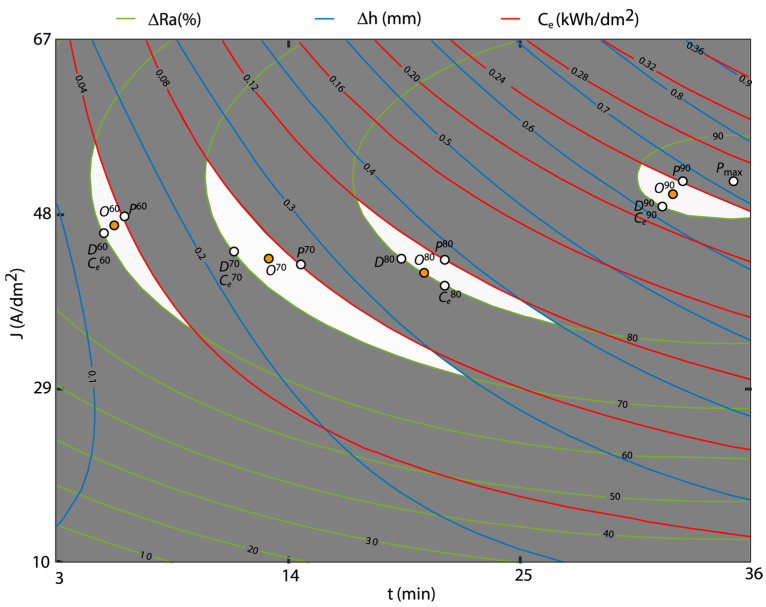
Overlaid contour plot for the initial texture $Ra_{1}^{0}$.

**Figure 12 materials-16-01770-f012:**
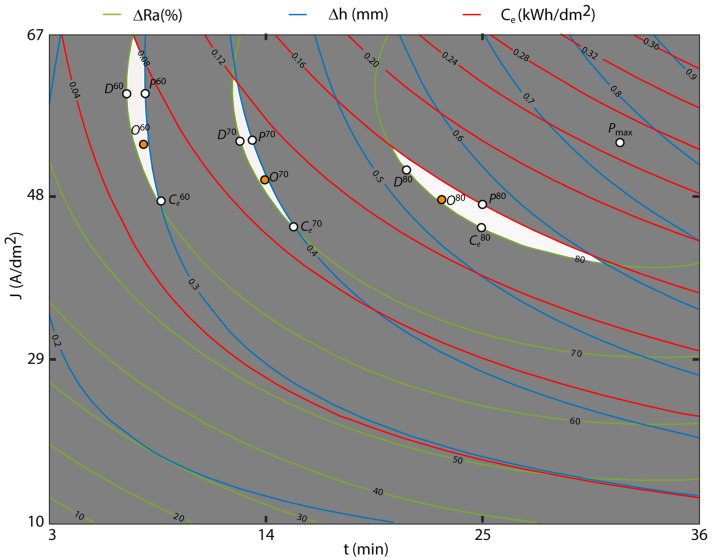
Overlaid contour plot for the initial texture $Ra_{2}^{0}$.

**Figure 13 materials-16-01770-f013:**
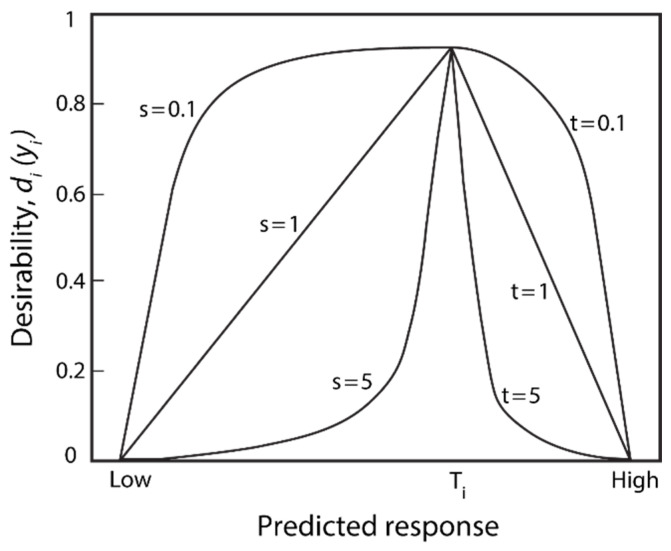
Form of the desirability function *d_i_*(*y_i_*) according to the values of the potentials *s* and *t*.

**Figure 14 materials-16-01770-f014:**
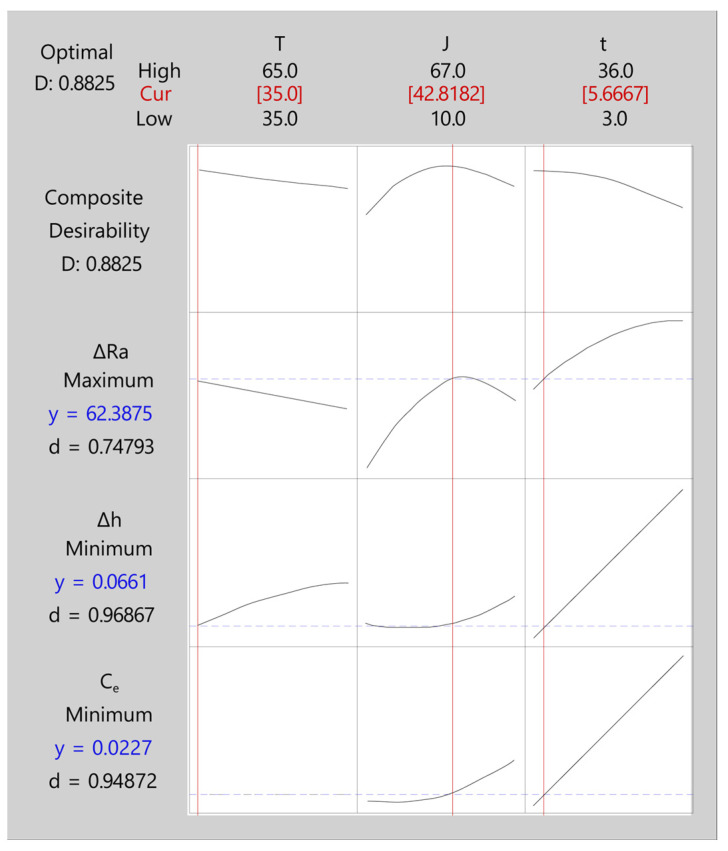
Multi-objective optimization based on the desirability function for the initial texture $Ra_{1}^{0}$.

**Figure 15 materials-16-01770-f015:**
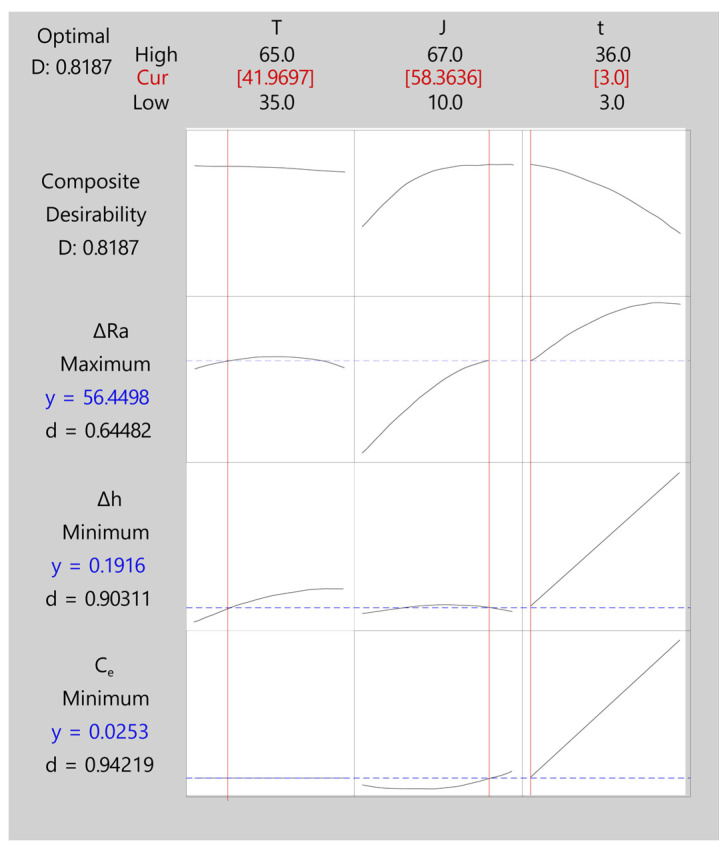
Multi-objective optimization based on the desirability function for initial texture $Ra_{2}^{0}$.

**Table 1 materials-16-01770-t001:** Factors and levels for the first factorial design (2^5^).

${Ra}_{i}^{0}(\mu m)$	*D_i_* (mm)	*T_i_* (°C)	*J_i_* (A/dm^2^)	*t_i_* (min)
0.5 ÷ 0.8	300	35	10	3
1.0 ÷ 1.3	150	65	67	36

**Table 2 materials-16-01770-t002:** Factors and levels in the 2^3^ × 2 factorial design.

${Ra}_{i}^{0}$ (µm)	*T_i_* (°C)	*J_i_* (A/dm^2^)	*t_i_* (min)
0.5 ÷ 0.8	35	10	3
	45	29	14
1.0 ÷ 1.3	55	48	25
	65	67	36

**Table 3 materials-16-01770-t003:** ANOVA analysis for the identification of significant factors on the EP process (DF = degrees of freedom; Adj SS = Adjusted sums of squares; Adj MS = Adjusted mean squares).

Source	DF	Adj SS (Type III)	Adj MS	F-Value	*p*-Value
Main effects					
A:D	1	314.641	314.641	2.91	0.0895
B:*Ra*^0^	1	1226.53	1226.53	11.35	0.0009
C:*T*	3	5434.81	1811.6	16.77	0.0000
D:*J*	3	116,275.0	38,758.2	358.69	0.0000
E:*t*	3	43,359.8	14,453.3	133.76	0.0000
Interactions					
A×B	1	4.25648	4.25648	0.04	0.8429
A×C	3	163.982	54.6607	0.51	0.6787
A×D	3	254.904	84.9681	0.79	0.5028
A×E	3	164.885	54.9616	0.51	0.6768
B×C	3	1201.83	400.609	3.71	0.0126
B×D	3	2139.52	713.173	6.60	0.0003
B×E	3	372.487	124.162	1.15	0.3305
C×D	9	14,686.0	1631.77	15.10	0.0000
C×E	9	739.626	82.1807	0.76	0.6530
D×E	9	5950.54	661.171	6.12	0.0000
Error	198	21,395.0	108.056		
Total	255	213,683.0			

**Table 4 materials-16-01770-t004:** ANOVA analysis of the response variable *ΔRa* for the initial texture $Ra_{1}^{0}$ (DF = degrees of freedom, Adj SS = Adjusted sums of squares, Adj MS = Adjusted mean squares).

Source	DF	Adj SS (Type III)	Adj MS	F-Value	*p*-Value
*J*	1	1859	1859.5	16.25	0.000
*t*	1	5583	5582.8	48.77	0.000
*T* ^2^	1	4889	4889.1	42.71	0.000
*J* ^2^	1	4964	4964.4	43.37	0.000
*t* ^2^	1	1560	1560.4	13.63	0.001
*T*×*J*	1	2824	2824.5	24.68	0.000
*J*×*t*	1	2032	2031.6	17.75	0.000
Error	56	6410	114.5		
Total	63	45,798			

**Table 5 materials-16-01770-t005:** Maximum polishing rates were obtained for the initial texture $Ra_{1}^{0}$.

Temperature (°C)	Maximum *ΔRa* (%)	EP Time (min)	Current Density (A/dm^2^)
35	89.98	35	48
45	90.72	35	52
55	89.34	35	56
65	85.85	35	60

**Table 6 materials-16-01770-t006:** ANOVA analysis of the response variable *ΔRa* for the initial texture $Ra_{2}^{0}$ (DF = degrees of freedom; Adj SS = Adjusted sums of squares; Adj MS = Adjusted mean squares).

Source	DF	Adj SS (Type III)	Adj MS	F-Value	*p*-Value
*J*	1	6084	6084.0	36.89	0.000
*t*	1	5496	5496.2	33.33	0.000
*J* ^2^	1	1903	1903.5	11.54	0.001
*t* ^2^	1	1499	1499.4	9.09	0.004
*J*×*t*	1	2124	2124.3	12.88	0.001
Error	57	9400	164.9		
Total	62	36,896			

**Table 7 materials-16-01770-t007:** ANOVA analysis of the response variable ${Ra}^{f}$ for the initial texture $Ra_{1}^{0}$ (DF = degrees of freedom, Adj SS = Adjusted sums of squares, Adj MS = Adjusted mean squares).

Source	DF	Adj SS (Type III)	Adj MS	F-Value	*p*-Value
*J*	1	0.08481	0.084811	15.74	0.000
*t*	1	0.14125	0.141251	26.21	0.000
*T* ^2^	1	0.15370	0.153703	28.52	0.000
*J* ^2^	1	0.23281	0.232806	43.19	0.000
*t* ^2^	1	0.03331	0.033306	6.18	0.016
*T*×*J*	1	0.10078	0.100779	18.70	0.000
*J*×*t*	1	0.04774	0.047742	8.86	0.004
Error	56	0.30183	0.005390		
Total	63	1.71170			

**Table 8 materials-16-01770-t008:** Final roughness ${Ra}^{f}$ obtained for the initial texture $Ra_{1}^{0}$.

Temperature (°C)	EP Time (min)	Current Density (A/dm^2^)	Minimum *Ra^f^* (µm)
35	35	46	0.035
45	35	52	0.049
55	35	56	0.051
65	35	62	0.044

**Table 9 materials-16-01770-t009:** ANOVA analysis of the response variable ${Ra}^{f}$ for the initial texture $Ra_{2}^{0}$ (DF = degrees of freedom, Adj SS = Adjusted sums of squares, Adj MS = Adjusted mean squares).

Source	DF	Adj SS (Type III)	Adj MS	F-Value	*p*-Value
*J*	1	0.6686	0.66861	31.37	0.000
*t*	1	0.6862	0.68621	32.20	0.000
*J* ^2^	1	0.1947	0.19470	9.13	0.004
*t* ^2^	1	0.1947	0.19470	9.13	0.004
*J*×*t*	1	0.2624	0.26240	12.31	0.001
Error	58	1.2362	0.02131		
Total	63	4.3777			

**Table 10 materials-16-01770-t010:** ANOVA analysis of the response variable *Δh* for the initial texture $Ra_{1}^{0}$ (DF = degrees of freedom; Adj SS = Adjusted sums of squares; Adj MS = Adjusted mean squares).

Source	DF	Adj SS (Type III)	Adj MS	F-Value	*p*-Value
*T*	1	0.19935	0.19935	24.77	0.000
*J*	1	0.03427	0.03427	4.26	0.043
*J* ^2^	1	0.03290	0.03290	4.09	0.048
*J*×*t*	1	2.58172	2.58172	320.78	0.000
Error	59	0.47484	0.00805		
Total	63	4.68603			

**Table 11 materials-16-01770-t011:** Minimum (*Δh_min_*) and maximum (*Δh_max_*) thickness variation for initial texture $Ra_{1}^{0}$.

Temperature (°C)	*Δh_min_* (%)	Current Density (A/dm^2^)	EP Time (min)	*Δh_max_* (%)	Current Density (A/dm^2^)	EP Time (min)
35	0.031	32	3	0.887	67	36
45	0.081	32	3	0.937	67	36
55	0.130	32	3	0.986	67	36
65	0.180	32	3	1.036	67	36

**Table 12 materials-16-01770-t012:** ANOVA analysis of the response variable *Δh* for the initial texture $Ra_{2}^{0}$ (DF = degrees of freedom, Adj SS = Adjusted sums of squares, Adj MS = Adjusted mean squares).

Source	DF	Adj SS (Type III)	Adj MS	F-Value	*p*-Value
*J* ^2^	1	0.09376	0.09376	8.58	0.005
*T*×*J*	1	0.12918	0.12918	11.82	0.001
*J*×*t*	1	2.31068	2.31068	211.37	0.000
Error	60	0.65592	0.01093		
Total	63	4.45284			

**Table 13 materials-16-01770-t013:** Minimum (*Δh_min_*) and maximum (*Δh_max_*) thickness variation for initial texture $Ra_{2}^{0}$.

Temperature (°C)	*Δh_min_* (%)	Current Density (A/dm^2^)	EP Time (min)	*Δh_max_* (%)	Current Density (A/dm^2^)	EP Time (min)
35	0.131	3	10	0.868	36	67
45	0.139	3	10	0.922	36	67
55	0.147	3	10	0.977	36	67
65	0.156	3	10	1.032	36	67

**Table 14 materials-16-01770-t014:** Optimum points per polishing percentage for initial texture $Ra_{1}^{0}$.

Electropolishing Range	Optimal Points	*J* (A/dm^2^)	*t* (min)	*P* (%)	*D* (mm)	*C* (kWh/dm^2^)
90%	*P* ^90^	52	33.0	90.50	0.680	0.240
	*D* ^90^	50	31.0	90.00	0.620	0.210
	*C* ^90^	50	31.0	90.00	0.620	0.210
	*O* ^90^	51	32.0	90.37	0.650	0.230
80%	*P* ^80^	44	21.0	82.06	0.380	0.120
	*D* ^80^	44	19.0	80.00	0.350	0.100
	*C* ^80^	40	21.0	80.00	0.350	0.100
	*O* ^80^	42	20.0	80.00	0.350	0.100
70%	*P* ^70^	42	15.0	74.13	0.271	0.080
	*D* ^70^	44	11.0	70.00	0.220	0.063
	*C* ^70^	44	11.0	70.00	0.220	0.063
	*O* ^70^	43	13.0	71.97	0.240	0.072
60%	*P* ^60^	48	6.3	62.50	0.150	0.040
	*D* ^60^	46	5.0	60.00	0.127	0.031
	*C* ^60^	46	5.0	60.00	0.127	0.031
	*O* ^60^	47	5.6	61.34	0.141	0.036

**Table 15 materials-16-01770-t015:** Optimum points per polishing percentage for the initial texture $Ra_{2}^{0}$.

Electropolishing Range	Optimal Points	*J* (A/dm^2^)	*t* (min)	*P* (%)	*D* (mm)	*C* (kWh/dm^2^)
80%	*P* ^80^	47.00	25.00	81.50	0.570	0.160
	*D* ^80^	51.50	21.00	80.00	0.531	0.156
	*C* ^80^	44.00	25.00	80.00	0.540	0.144
	*O* ^80^	47.75	23.00	80.47	0.539	0.150
70%	*P* ^70^	56.00	13.50	70.80	0.400	0.110
	*D* ^70^	56.00	12.50	70.00	0.390	0.107
	*C* ^70^	44.00	15.50	70.00	0.400	0.088
	*O* ^70^	50.00	14.00	70.70	0.390	0.098
60%	*P* ^60^	60.00	7.50	62.00	0.300	0.073
	*D* ^60^	60.00	7.00	60.00	0.280	0.064
	*C* ^60^	47.50	8.50	60.00	0.300	0.056
	*O* ^60^	53.75	7.75	60.82	0.289	0.061

**Table 16 materials-16-01770-t016:** Optimization EP parameters for the initial texture $Ra_{1}^{0}$.

Response	Goal	Lower	Target	Upper	Weight	Importance
*ΔRa*	Maximum	0.0000	83.4133	83.4133	1	1
*Δh*	Minimum	0.0300	0.0300	1.18100	1	1
*C_e_*	Minimum	0.0025	0.0025	0.39799	1	1

**Table 17 materials-16-01770-t017:** The five best results obtained with the desirability function for the initial texture $Ra_{1}^{0}$.

Solution	*T*	*J*	*t*	*ΔRa* Fit	*Δh* Fit	*C_e_* Fit	Composite Desirability
1	35.0000	42.8182	5.66667	62.3875	0.066059	0.0227400	0.882521
2	35.0008	51.4402	3.00106	57.2971	0.039154	0.0151337	0.870483
3	35.0123	52.1628	3.00127	57.1020	0.040973	0.0160720	0.868321
4	35.0015	52.8449	3.00146	56.8909	0.042548	0.0169848	0.866159
5	49.9811	48.7552	3.00000	53.3307	0.140660	0.0119065	0.826252

**Table 18 materials-16-01770-t018:** EP conditions were obtained for multi-response optimization for the initial texture $Ra_{1}^{0}$.

Response	Fit	SE Fit	95% CI
*ΔRa*	62.3875	3.34 (4%)	(55.71; 69.07)
*Δh*	0.066059	0.0289 (2.44%)	(0.0082; 0.1239)
*C_e_*	0.02274	0.00164 (0.41%)	(0.01945; 0.02603)

**Table 19 materials-16-01770-t019:** Optimization EP parameters for the initial texture $Ra_{2}^{0}$.

Response	Goal	Lower	Target	Upper	Weight	Importance
*ΔRa*	Maximum	0.0000	87.5433	87.5433	1	1
*Δh*	Minimum	0.0930	0.0930	1.11100	1	1
*C_e_*	Minimum	0.0025	0.0025	0.39799	1	1

**Table 20 materials-16-01770-t020:** The five best results obtained with the desirability function for the initial texture $Ra_{2}^{0}$.

Solution	*T*	*J*	*t*	*ΔRa* Fit	*Δh* Fit	*C_e_* Fit	Composite Desirability
1	41.9697	58.3636	3	56.4498	0.191636	0.0253191	0.818665
2	36.7169	66.9963	3	52.3271	0.117671	0.0418509	0.806792
3	35.1342	66.9962	3	50.7346	0.098369	0.0418508	0.803661
4	35.1003	66.9962	3	50.6988	0.097945	0.0418508	0.803584
5	35.0971	66.9962	3	50.6954	0.097905	0.0418508	0.803577

**Table 21 materials-16-01770-t021:** EP conditions obtained for multi-response optimization for the initial texture $Ra_{2}^{0}$.

Response	Fit	SE Fit	95% CI
*ΔRa*	56.45	4.88 (5.57%)	(46.66; 66.23)
*Δh*	0.1916	0.0335 (3.01%)	(0.1245; 0.2588)
*C_e_*	0.02532	0.00194 (0.48%)	(0.02144; 0.02920)

## Data Availability

The data presented in this study are available upon request from the corresponding author.

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
