# Peer review of "Electropolishing Stainless Steel Optimization Using Surface Quality, Dimensional Accuracy, and Electrical Consumption Criteria"

_materials, 2023, doi:10.3390/ma16051770_

Round 1

Reviewer 1 Report

1) The article presents an interesting research topic. Please consider reviewing the abstract and highlight the novelty, major findingsandconclusions.

2) The results are merely described and is limited to comparing the experimental observation. The authors are encouraged to include a discussionsection and critically discuss more the observations from this investigation with existing literature.

3) Suggestions of literature:

a) book, Nonconventional Machining, DE Gruyter, 2023, ISBN: 9783110584103

b) book, Nontraditional Machining, Springer, 2013, ISBN: 978-1-4471-5178-4

b) book, Computational Methods and Production Engineering, Elsevier, 2017, ISBN: 9780857094810h

c) book, Statistical and Computational Techniques in Manufacturing, Springer, 2012, ISBN: 978-3-642-25859-6

d) book, Design of Experiments in Production Engineering. Springer, 2016, ISBN 978-3-319-23837-1

and others articles of reputed international journals

4) Improve the conclusions

Reviewer 2 Report

Over all the paper is well written, the findings of the research are good. Address the following queries to make it look better.

1.       In the abstract, add the key findings and their influence on the study (may not be the numerical values, but the general discussions).

2.       I find the introduction is too vast and most of the basics related to electropolishing is discussed. For a technical paper you can limit it to one page and discuss only the necessary information related to your manuscript.

3.       The discussion written wrt. Figure 3 is very less. The authors are suggested to elaborate the significance of each plot in this figure and discuss how it is influencing the study.

4.       In table 3 P-value for one main effect and few interaction effects are higher than 0.05, that means the selected factors are not correct. The explanation regarding this is required.

5.       In page no. 9, while explaining about the figure 4 at some place the author mentions Fig. 49!! what is that??

6.       General discussion is given about figure 4. The author should explain the significance, the meaning of the colour contours in the figure.

7.       Equations 17-19 usage is not very clear in the manuscript, kindly rewrite.

8.       What does the curved lines and slopes indicate in the figure 15 and 16?

9.       Entire work is based on the design of experiments, the author should show some, before and after images of the specimens.

10.   If the images are not available in the results section, the authors can discuss some similar studies in the results section.  

Reviewer 3 Report

- What is the most important innovation of this article?

- Authors should provide explanations for each of the references given in the introduction section.

- What was the reason for using the response surface method compared to other design of experiment methods?

- How are the upper and lower limits provided for each parameter determined? Did the authors use the trial and error method, or did the selection of the parameter range have another scientific basis?

- The interpretations provided for the parameter interaction results are very incomplete and require more complete explanations.

- Why did the authors not use methods such as Sobol sensitivity analysis?

- How have the efficiency and performance of regression equations been investigated?

Round 2

Reviewer 3 Report

The manuscript can be accepted in the present form.